# Augmentation of Bri2 molecular chaperone activity against amyloid-β reduces neurotoxicity in mouse hippocampus in vitro

Gefei Chen[1], Yuniesky Andrade-Talavera [2], Simone Tambaro[1], Axel Leppert [1], Harriet E. Nilsson[3], Xueying Zhong[3], Michael Landreh [4], Per Nilsson [1], Hans Hebert [3], Henrik Biverstål [1,5], André Fisahn[2], Axel Abelein [1] & Jan Johansson[1]*

Molecular chaperones play important roles in preventing protein misfolding and its potentially harmful consequences. Deterioration of molecular chaperone systems upon ageing are thought to underlie age-related neurodegenerative diseases, and augmenting their activities could have therapeutic potential. The dementia relevant domain BRICHOS from the Bri2 protein shows qualitatively different chaperone activities depending on quaternary structure, and assembly of monomers into high-molecular weight oligomers reduces the ability to prevent neurotoxicity induced by the Alzheimer-associated amyloid-β peptide 1-42 (Aβ42). Here we design a Bri2 BRICHOS mutant (R221E) that forms stable monomers and selectively blocks a main source of toxic species during Aβ42 aggregation. Wild type Bri2 BRICHOS oligomers are partly disassembled into monomers in the presence of the R221E mutant, which leads to potentiated ability to prevent Aβ42 toxicity to neuronal network activity. These results suggest that the activity of endogenous molecular chaperones may be modulated to enhance anti-Aβ42 neurotoxic effects.

[1] Department of Neurobiology, Care Sciences and Society, Center for Alzheimer Research, Division of Neurogeriatrics, Karolinska Institutet, 141 57 Huddinge, Sweden. [2] Department of Neurobiology, Care Sciences and Society, Center for Alzheimer Research, Division of Neurogeriatrics, Neuronal Oscillations Laboratory, Karolinska Institutet, 171 77 Stockholm, Sweden. [3] School of Engineering Sciences in Chemistry, Biotechnology and Health, Department of Biomedical Engineering and Health Systems, KTH Royal Institute of Technology, Department of Biosciences and Nutrition, Karolinska Institutet, 141 52 Huddinge, Sweden. [4] Science for Life Laboratory, Department of Microbiology, Tumour and Cell Biology, Karolinska Institutet, Tomtebodavägen 23A, 171 65 Stockholm, Sweden. [5] Department of Physical Organic Chemistry, Latvian Institute of Organic Synthesis, Aizkraukles 21, Riga LV-1006, Latvia. *email: janne.johansson@ki.se

Molecular chaperones are ubiquitously expressed and responsible for the maintenance of protein homeostasis by transiently binding their substrates to assist them in folding and trafficking, and to prevent them from aggregating and exerting cytotoxic effects[1]. Surveillance by human molecular chaperones may decline during aging, resulting in increased cellular stress and eventually leading to neurodegenerative diseases and other disorders[2–4]. Thus, pharmacologically augmenting or stimulating chaperone capacity could have therapeutic potential[4–6], and strategies to increase the specificity and activity of the interactions of molecular chaperones with their target substrates are sought for[7,8].

Proteins can misfold and self-assemble into highly ordered fibrillar amyloid structures with toxic effects, a phenomenon that is linked to ~40 human diseases, including type-2 diabetes, the neurodegenerative disorders Parkinson disease, and Alzheimer's disease (AD)[9,10]. AD is the most common form of dementia and is developing into a global challenge since no disease-modifying treatment exists. The development of AD treatments has been outstandingly unsuccessful, with a failure rate above 99% in clinical trials[11]. Several observations support that amyloid-β peptide (Aβ) aggregation contributes to the pathogenesis of AD. The Alzheimer plaques are composed of Aβ, mutations in the Aβ precursor protein (AβPP) that increase the production or augment aggregation propensity of Aβ lead to early onset AD and to AD-like phenotypes in animal models. In contrast, a mutation that lowers Aβ production is protective[12–14]. Proteolytic processing of AβPP generates Aβ peptides of different lengths, whereof Aβ42 is the most aggregation prone and toxic[15]. The kinetics of Aβ42 aggregation follows recently defined nucleation-dependent microscopic events[16]. During primary nucleation Aβ42 monomers associate and form a nucleus, from which a fibril can start to elongate. During secondary nucleation, monomers attach to the surface of a fibril, which catalyses the formation of a new nucleus, leading to exponential fibril growth. This monomer-dependent secondary nucleation autocatalytic pathway is the predominant source of toxic Aβ42 species[17]. Key challenges in finding a treatment of AD is to specifically reduce Aβ42 neurotoxicity, rather than focusing on overall aggregation and plaque formation[18,19]. Many compounds have been suggested to interfere with amyloid formation[18], including molecular chaperones[20–23] as well as chaperone-like proteins[24–27]. The effects of chaperones on Aβ42 fibrillization are diverse when it comes to effects on different nucleation events; DNAJB6 from the heat shock protein 40 (Hsp40) family for example prevents predominately primary nucleation[23], while αB-crystallin and calcium-binding protein nuclebindin-1 interfere with both elongation and secondary nucleation events[23,25].

The BRICHOS domain has been found in several human precursor proteins, initially in Bri2, chondromodulin-1, and prosurfactant protein C (proSP-C)[28,29]. The BRICHOS domain has been suggested to prevent amyloid formation of aggregation prone regions (clients) of the respective proprotein during biosynthesis[30–32]. Recombinant human (rh) BRICHOS domains from proSP-C and Bri2 are efficient inhibitors also of amyloid formation of non-client proteins, such as medin, islet amyloid polypeptide (IAPP), Aβ40, and Aβ42[33–35]. Bri2, mutations of which give rise to amyloid formation and familial British and Danish dementias, is produced in the central nervous system (CNS), with expression in neurons of the hippocampus and cortex in humans[36,37], and colocalizes with senile plaques in AD patients[38,39]. Under physiological conditions, the BRICHOS domain is released by proteolysis from the Bri2 precursor protein[33,38,40]. Rh Bri2 BRICHOS is efficient in inhibiting Aβ42 fibril formation in vitro and in alleviating the related neurotoxicity to hippocampal slice preparations and in *Drosophila* models[26,27,34].

Rh proSP-C BRICHOS specifically impedes the secondary nucleation step in Aβ42 fibril formation[19]. Rh Bri2 BRICHOS modulates both elongation and secondary nucleation events, but different assembly states of Bri2 BRICHOS affect Aβ fibril formation in different ways[23,26,27]. Bri2 BRICHOS monomers are most potent in preventing Aβ42-induced disruption of neuronal network activity, while dimers most efficiently suppress Aβ42 overall fibril formation and oligomers inhibit non-fibrillar protein aggregation[26]. The Bri2 BRICHOS monomers are not long-term stable and form high-molecular weight oligomers in a concentration-dependent manner in phosphate buffer or in mouse serum in vitro, which is accompanied by reduced potency against Aβ42 fibril formation[26]. Conversion of Bri2 BRICHOS monomers to high-molecular weight oligomers may be relevant for AD, as increased amounts of different Bri2 forms were found in AD brain compared with healthy controls[38]. These observations imply that modulating the distribution of Bri2 BRICHOS assembly states so that the amount of monomers is increased is a concept to combat Aβ42 neurotoxicity.

Here, we design a single point mutant of rh Bri2 BRICHOS that stabilizes the monomeric state. This mutant monomer is potent in preventing Aβ42 neurotoxicity, specifically suppresses secondary nucleation during fibril formation and, importantly, it potentiates wild-type protein against Aβ42 neurotoxicity.

## Results

**R221E mutant forms stable monomers and unstable oligomers.** The crystal structure of rh proSP-C BRICHOS[30], the only available high-resolution structure of a BRICHOS domain, shows a homotrimer in which residues from helix 2 point into a pocket of the neighbouring subunit (Supplementary Fig. 1a). In a structural model of Bri2 BRICHOS subunit based on the proSP-C BRICHOS structure (Fig. 1a)[31,34] Arg221 is surface exposed in helix 2 and can point into the pocket of a neighbouring subunit. This motivated the Arg221Glu mutation, to introduce opposite surface electrostatic potential (Fig. 1b, Supplementary Fig. 1b, c), with the aim to destabilize the oligomer and generate a stable subunit monomer.

Rh Bri2 BRICHOS R221E was produced in fusion with a solubility tag, NT*, that was recently developed based on the N-terminal domain of spider silk proteins[26,41,42]. Purified NT*-Bri2 BRICHOS R221E was separated into oligomers, dimers, and monomers by size-exclusion chromatography (SEC; Fig. 1c). In contrast to wild-type protein, the mutant forms to a large extent monomers (Fig. 1c), suggesting that Arg221 indeed contributes to Bri2 BRICHOS oligomerization. After proteolytic release of the NT* tag, isolated rh Bri2 BRICHOS R221E oligomers are partially linked by intersubunit disulfide bonds, and the dimers are disulfide dependent (Supplementary Fig. 2a, b). Electrospray ionization mass spectrometry (ESI-MS) confirmed the quaternary structure of rh Bri2 BRICHOS R221E monomers isolated by SEC (Supplementary Fig. 3a). The mass determined by ESI-MS, 14,050.2 Da, is in perfect agreement with the calculated mass, 14,050.1 Da, of a monomer in which the two conserved Cys form an intramolecular disulfide bond. Circular dichroism spectroscopy showed that the overall secondary structure of monomeric rh Bri2 BRICHOS R221E is similar to wild-type monomers (Supplementary Fig. 3b). In association with the formation of oligomers, the negative circular dichroism peak ~205–210 nm shifts to the right (Supplementary Fig. 3b), indicative of structural stabilization which is comparable to the pattern observed for rh wild-type Bri2 BRICHOS[26]. Isolated rh Bri2 BRICHOS R221E monomers, dimers, and oligomers bind bis-ANS, evidenced by a marked increase in fluorescence emission intensity compared with free bis-ANS, and a blue shift of the emission maximum

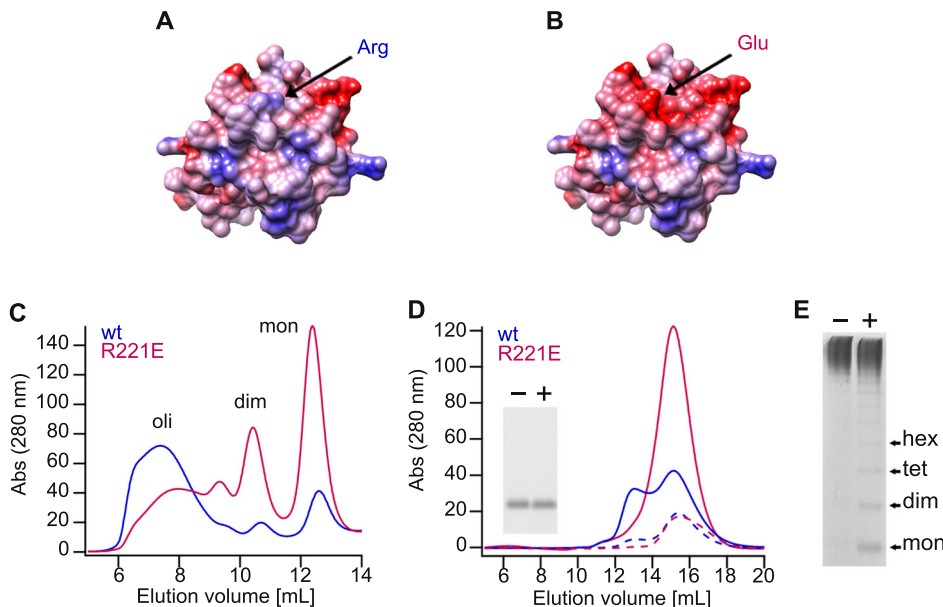

**Fig. 1 Rh Bri2 BRICHOS R221E forms stable monomers and unstable oligomers. a, b** Homology models of wild-type Bri2 BRICHOS **a** based on the proSP-C structure[27,34] and Bri2 BRICHOS R221E **b** rendered with red for negative surface electrostatic potential ($-10$ kcal mol$^{-1}$), light pink for near neutral, and blue for positive potential ($10$ kcal mol$^{-1}$). The arrows point to Arg221 for wild-type Bri2 BRICHOS and Glu221 for Bri2 BRICHOS R221E. **c** SEC of wild type (wt) NT*-Bri2 BRICHOS and NT*-Bri2 BRICHOS R221E. Oli, oligomers; dim, dimers; mon, monomers. **d** SEC of isolated monomers of rh Bri2 BRICHOS R221E and rh wt Bri2 BRICHOS at low concentration (~7.5 μM, dashed curves) and high concentrations (54 μM for R221E and 30 μM for wt, solid curves). The inset shows native PAGE of rh Bri2 BRICHOS R221E monomers before (−) and after (+) overnight incubation at 37 °C in 20 mM NaPi pH 8.0. **e** Rh Bri2 BRICHOS R221E oligomers analysed by native PAGE before (−) and after (+) overnight incubation at 37 °C in 20 mM NaPi pH 8.0. Hex, hexamers; tet, tetramers; dim, dimers; mon, monomers.

from ~533 nm to 480–490 nm (Supplementary Fig. 3c). This shows that all rh Bri2 BRICHOS R221E species expose hydrophobic surfaces, which is similar to the situation for the wild-type species[26]. The abilities of different assembly states of rh Bri2 BRICHOS R221E to prevent non-fibrillar protein aggregation were assessed against thermo-denatured citrate synthase. The dimers and monomers were comparatively inefficient in suppressing citrate synthase aggregation, while the oligomers efficiently reduce aggregation of citrate synthase (Supplementary Fig. 3d), like the case for wild-type species. Throughout this paper the concentration of different rh Bri2 BRICHOS R221E species is referred to its monomeric subunits.

Isolated rh wild-type Bri2 BRICHOS monomers form dimers upon increasing concentration as seen by SEC, while rh Bri2 BRICHOS R221E monomers did not show altered migration on SEC at increasing concentration (Fig. 1d). Likewise, incubation of rh Bri2 BRICHOS R221E monomers in 20 mM NaPi pH 8.0 at 37 °C overnight followed by native polyacrylamide gel electrophoresis (PAGE) showed that they maintain a monomeric state (Fig. 1d, inset), which is not the case for the wild-type protein[26]. Analysis by native PAGE shows that monomers and other low-n species are released from rh Bri2 BRICHOS R221E oligomers after overnight incubation at 37 °C (Fig. 1e), while rh wild-type Bri2 BRICHOS oligomers do not spontaneously release smaller species during incubation (Fig. 2, see further below).

**Oligomer destabilization augments Bri2 BRICHOS activity.** To investigate the effects of destabilization of rh Bri2 BRICHOS oligomers on alleviating Aβ42-associated neurotoxicity, we tested the efficacies in preventing Aβ42-induced reduction of γ oscillations in mouse hippocampal slices. Gamma oscillations correlate with learning, memory, cognition and other higher processes[43,44], and cognitive decline observed in AD patients goes in-hand with

a decrease of γ oscillations[45–47]. Here, γ oscillations were induced in horizontal hippocampal slices from wild-type C57BL/6 mice by superfusing slices with 100 nM kainic acid and then allowing them to stabilize for 30 min. Preincubation of hippocampal slices with 50 nM Aβ42 for 15 min markedly reduced the power of γ oscillations generated by subsequent kainic acid application (Fig. 2a–d, Supplementary Data 1). Addition of 50 nM rh BRICHOS R221E oligomers together with 50 nM Aβ42 resulted in γ oscillations that did not differ from those in non-treated controls, while addition of rh wild-type Bri2 BRICHOS oligomers with 50 nM Aβ42 resulted in γ oscillations that were lower compared to the non-treated controls and to the γ oscillations obtained after treatment with rh Bri2 BRICHOS R221E oligomers and Aβ42 (Fig. 2a–d). These observations suggest that destabilization of rh Bri2 BRICHOS R221E oligomers and release of smaller species (Fig. 1e) increases the anti-Aβ42 neurotoxicity activity. We next hypothesized that rh Bri2 BRICHOS R221E monomers can interfere with rh wild-type Bri2 BRICHOS oligomer formation and thereby increase the ratio of monomers, which would be expected to increase the potency to counteract Aβ42 neurotoxicity.

To test this hypothesis, we coincubated rh Bri2 BRICHOS R221E monomers and rh wild-type Bri2 BRICHOS oligomers, after which the distribution of Bri2 BRICHOS assembly states, and effects on Aβ42 fibril formation and neurotoxicity were evaluated. To enable differentiation between wild-type and R221E rh Bri2 BRICHOS monomers, we used a rh wild-type Bri2 BRICHOS construct containing an AU1 tag, which behaves like the non-tagged wild-type protein and can be selectively immunodetected using an anti-AU1 antibody[48]. Native PAGE and western blot analysis showed that coincubation of rh wild-type Bri2 BRICHOS oligomers with rh Bri2 BRICHOS R221E monomer lead to the release of smaller species, especially monomers (Fig. 2e). We used a fusion of Bri2 BRICHOS with

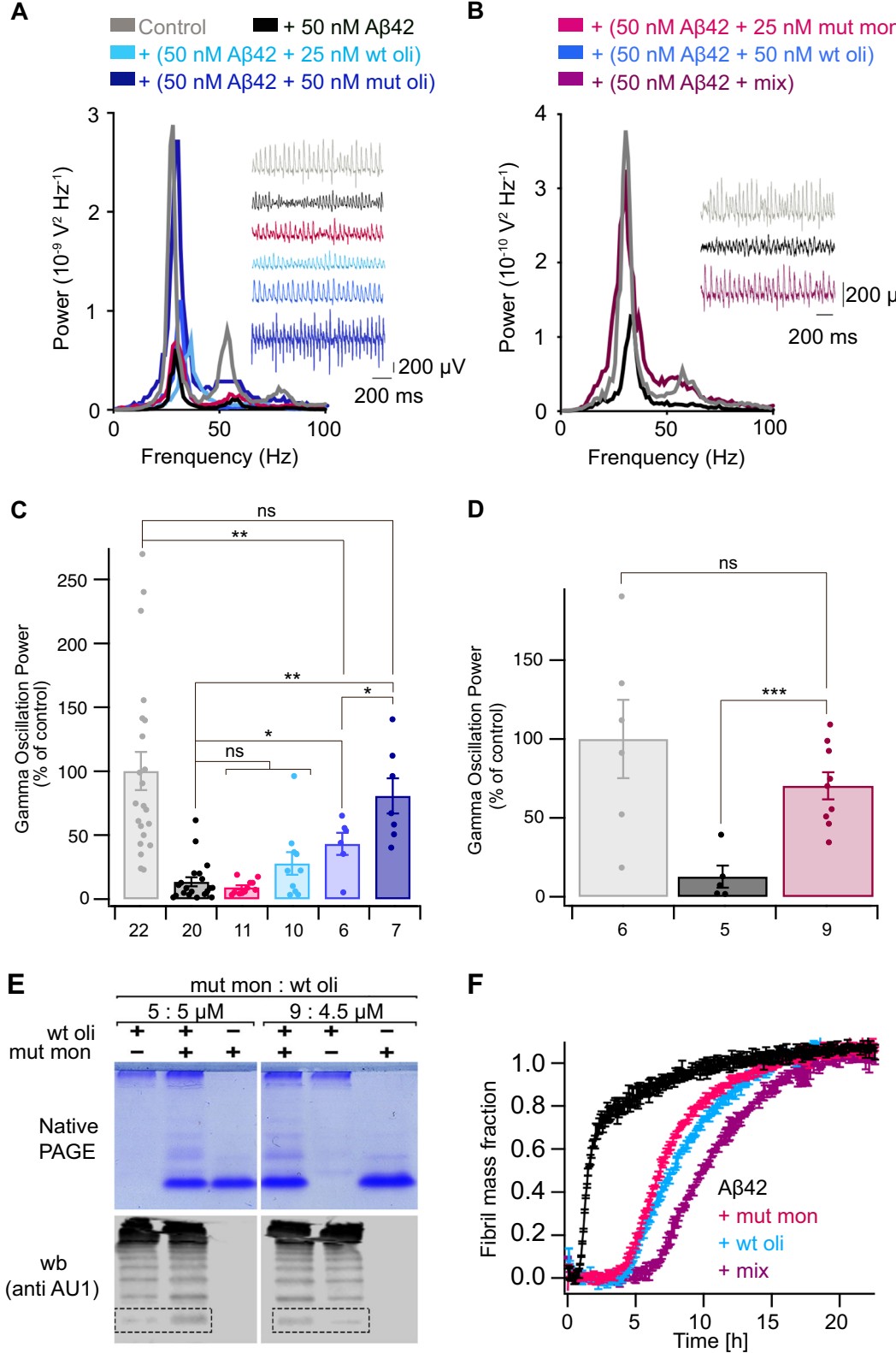

mCherry to further study the ability of rh Bri2 BRICHOS R221E to release monomers from wild-type rh Bri2 BRICHOS oligomers. Coincubation of rh Bri2 BRICHOS R221E monomers and rh Bri2 BRICHOS-mCherry oligomers gives a concentration-dependent release of wild-type monomers that is saturated at ~2:1 molar ratio between mutant and wild-type oligomers (Supplementary Fig. 5). Addition of rh Bri2 BRICHOS R221E monomer

also increases the anti-Aβ42 neurotoxic efficacy of rh wild-type Bri2 BRICHOS (Fig. 2a–d). We coincubated 25 nM of rh wild-type Bri2 BRICHOS oligomers and 25 nM of rh Bri2 BRICHOS R221E monomer overnight at 37 °C and the mixture attenuated Aβ42-induced reduction of γ oscillation power more potently than 50 nM of rh wild-type Bri2 BRICHOS oligomers preincubated in the same way as the mixture (Fig. 2a–d). Notably, the

**Fig. 2 Rh Bri2 BRICHOS R221E monomers potentiate wild-type Bri2 BRICHOS oligomers against Aβ42 neurotoxicity. a, b** Example traces and example power spectra under control conditions (gray), after 15 min incubation with 50 nM Aβ42 (black), 50 nM Aβ42 + 25 nM preincubated rh Bri2 BRICHOS R221E monomer (red), 50 nM Aβ42 + 25 nM preincubated rh wild-type (wt) Bri2 BRICHOS oligomer (light blue), 50 nM Aβ42 + 50 nM preincubated rh wt Bri2 BRICHOS oligomer (blue), 50 nM Aβ42 + 50 nM preincubated rh Bri2 BRICHOS R221E oligomer (dark blue), and 50 nM Aβ42 + a preincubated mixture of 25 nM rh wt Bri2 BRICHOS oligomer and 25 nM rh Bri2 BRICHOS R221E monomer (purple). Summary dot plot of normalized γ oscillation power **c, d** from the experiments are shown with the same colour coding as in **a** and **b**. The numbers under the histograms denote the number of biological replicates, and the data are reported as means ± standard errors of the means. Control vs. Aβ42: $p < 0.0001$, ns, no significant difference, *$p < 0.05$, **$p < 0.01$, ***$p < 0.001$. **e** Rh Bri2 BRICHOS R221E monomer and rh wt Bri2 BRICHOS oligomers containing an AU1 tag for immunodetection (5:5 and 9:4.5 µM) were coincubated at 37 °C overnight, and analysed by native PAGE and western blot with an anti-AU1 tag antibody. The dashed boxes indicate migration of wt Bri2 BRICHOS monomers. The results for control samples are shown in Supplementary Fig. 4. **f** ThT kinetic analysis of 3 µM Aβ42 alone (black), in the presence of 3 µM rh wt Bri2 BRICHOS oligomer (light blue), 3 µM rh Bri2 BRICHOS R221E monomer (red), or a preincubated mixture of 1.5 µM wt oligomer and 1.5 µM R221E monomer (purple).

enhanced efficiency of the mixture cannot be explained by simply adding the efficiencies of the two single species (Fig. 2a–d) and Aβ42 fibrillization kinetics were more strongly delayed by the incubated mixture of rh Bri2 BRICHOS R221E monomers and rh wild-type Bri2 BRICHOS oligomers than by the same total concentration of either species alone (Fig. 2f). Hence, these results suggest that shifting the distribution of Bri2 BRICHOS from oligomers to monomers is a way to potentiate the anti-Aβ42 neurotoxic capacity.

**R221E monomers selectively inhibit secondary nucleation.** To find out the molecular mechanism underlying the release of small species from the rh wild-type Bri2 BRICHOS oligomers and how this correlates to the increased capacity against Aβ42 neurotoxicity, we investigated the effects of rh Bri2 BRICHOS R221E monomers and dimers on Aβ42 fibril formation. We used thio-flavin ThT fluorescence[49] to monitor kinetics of Aβ42 fibril formation in the absence and presence of different concentrations of rh Bri2 BRICHOS R221E species (Fig. 3). The rh Bri2 BRICHOS R221E monomer showed a dose-dependent progressive reduction of Aβ42 fibril formation at substoichiometric concentrations, and the aggregation kinetics follows a typical sigmoidal behaviour (Fig. 3a). The fibrillization half time, $\tau_{1/2}$, increases with increasing rh Bri2 BRICHOS R221E monomer concentration, while the maximum rate of aggregation, $r_{max}$, shows a mono-exponential decline (Fig. 3a, c, Supplementary Data 2). The rh Bri2 BRICHOS dimers show similar effects as the monomers, but are more efficient in supressing the overall rate of Aβ42 fibril formation (Fig. 3b, c, Supplementary Data 2). The γ-value, $\tau_{1/2} \propto m(0)^{\gamma}$, where $m(0)$ is the initial Aβ42 monomer concentration, is similar in the absence and presence of both rh Bri2 BRICHOS R221E species (Supplementary Fig. 6a–d), suggesting that in the presence of rh Bri2 BRICHOS R221E, Aβ42 fibrillization still follows mainly monomer-dependent secondary pathways. The final ThT fluorescence intensity, which is an indicator of the mass of mature fibrils, shows a linear increase against Aβ42 concentration (Supplementary Fig. 7a), and does not change in the presence of different concentrations of rh Bri2 BRICHOS R221E species (Supplementary Fig. 7a, b).

Aβ42 fibrillization kinetics are described by a set of micro-scopic rate constants, i.e., for primary ($k_n$) and secondary (monomer-dependent $k_2$ and monomer-independent $k_-$) nucleation as well as elongation ($k_+$)[50,51]. To identify microscopic processes in the fibrillization pathways of Aβ42 that are most affected by different rh Bri2 BRICHOS R221E species, we determined the combined rate constants $\sqrt{k_n k_+}$ for primary and $\sqrt{k_+ k_2}$ for secondary nucleation events, respectively[51–53]. We fitted the kinetic model globally at different Aβ42 concentrations and a constant rh Bri2 BRICHOS R221E concentration (Supplementary Fig. 6a–c), where $\sqrt{k_n k_+}$ and

$\sqrt{k_+ k_2}$ are constrained to the same value across all concentrations. We found that the fitting parameter $\sqrt{k_n k_+}$ is similar in the presence of either rh Bri2 BRICHOS R221E monomer or dimer, as for Aβ42 alone (Supplementary Fig. 6e), suggesting that mainly secondary pathways are modulated by rh Bri2 BRICHOS R221E. The results also show that the rh Bri2 BRICHOS R221E dimer is more efficient than the monomer in inhibiting nucleation events related to $k_+ k_2$, i.e., fibril elongation or/and secondary nucleation. To quantitatively elucidate the effects of rh Bri2 BRICHOS R221E on Aβ42 fibrillization, we subsequently fitted the kinetic model individually to a dataset obtained with a constant Aβ42 concentration and different rh Bri2 BRICHOS R221E concentrations (Fig. 3a–c). The analysis revealed that $\sqrt{k_+ k_2}$ was changed by rh Bri2 BRICHOS R221E while $\sqrt{k_n k_+}$ was not (Fig. 3a, b, insets), again suggesting that the secondary nucleation is the main pathway affected. The individual fitting also showed that the dimers are more potent in reducing $\sqrt{k_+ k_2}$ than the monomer (Fig. 3a, b, insets).

Perturbations of the individual microscopic rate constants $k_+$, $k_n$, and $k_2$, are most relevant since their relative contributions determine the number of newly formed nucleation units, which might be linked to the generation of neurotoxic Aβ42 oligomeric species[18,19]. To evaluate the effects on individual microscopic processes, we performed global fits of the kinetic data set at constant Aβ42 and different rh Bri2 BRICHOS R221E concentrations, where the fits were constrained such that only one single rate constant, i.e., $k_n$, $k_+$, or $k_2$, is the sole fitting parameter (Supplementary Fig. 8). The primary nucleation rate $k_n$ or elongation $k_+$ as the sole fitting parameter gave rise to insufficient fits for both rh Bri2 BRICHOS species, which is further confirmed by nuclear magnetic resonance (NMR) data showing that neither monomers nor dimers of rh Bri2 BRICHOS R221E interact with Aβ40 monomers (Supplementary Fig. 9). With $k_2$ as the sole free fitting parameter the kinetics were adequately described for rh Bri2 BRICHOS R221E monomers, but not for the dimers (Supplementary Fig. 8). We conclude hence that the rh Bri2 BRICHOS R221E monomer mainly inhibits the secondary nucleation during Aβ42 fibril formation. In contrast, the dimers inhibit both secondary nucleation ($k_2$) and fibril-end elongation ($k_+$).

To further study how the elongation process is affected in the presence of different rh Bri2 BRICHOS species, we determined aggregation kinetics in the presence of a high initial fibril seed concentration, conditions under which the primary and second-ary nucleation events are negligible, and only fibril elongation contribute to the increase in the fibril mass[50]. Under these conditions, the fibrillization traces typically follow a concave aggregation behaviour (Fig. 3d, e, Supplementary Data 3), where the initial slope is directly proportional to the elongation rate $k_+$[50]. These seeding experiments revealed that the rh Bri2 BRICHOS R221E dimer decreases the elongation rate in a dose-

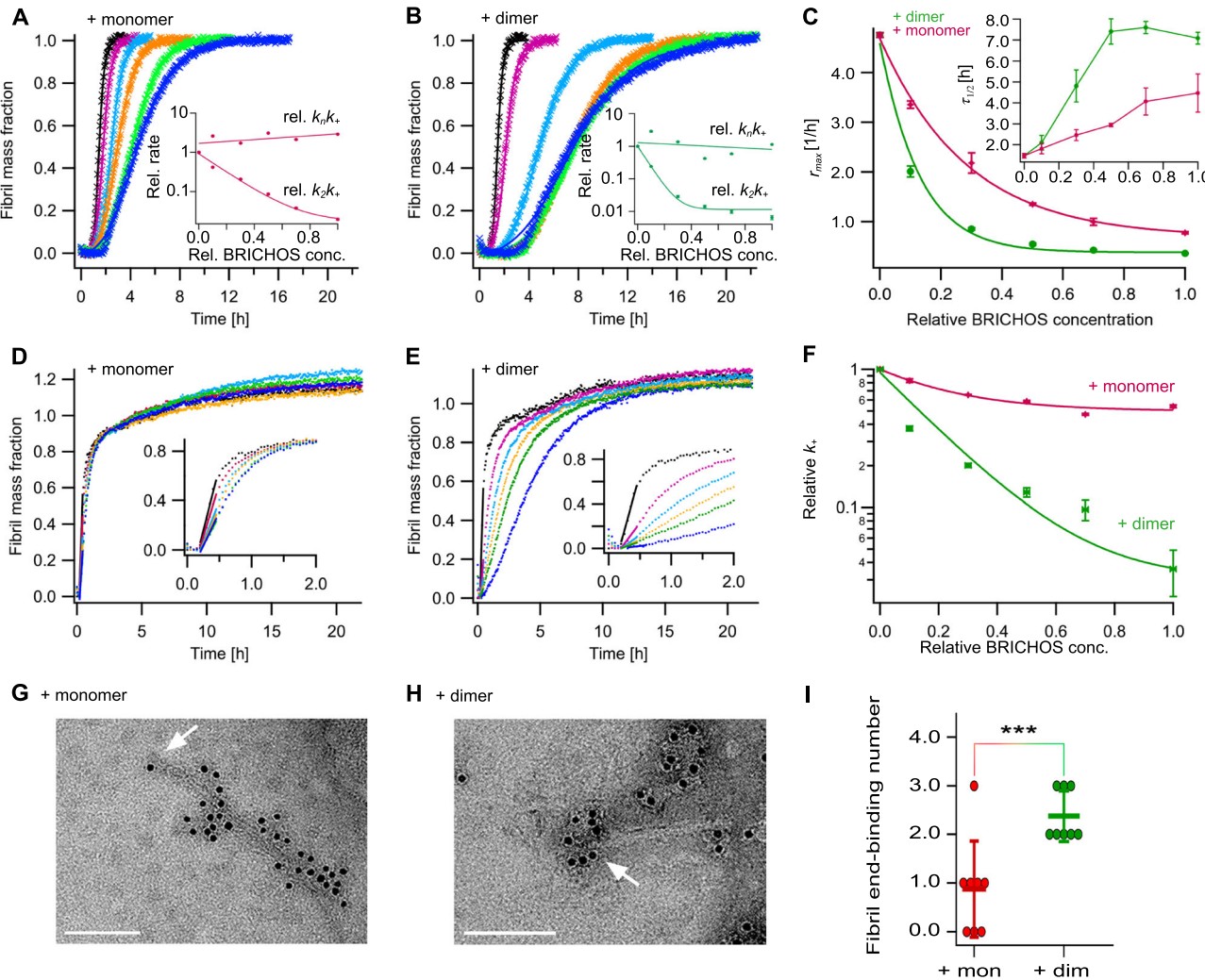

**Fig. 3 Effect of rh Bri2 BRICHOS R221E monomers and dimers on the microscopic events of Aβ42 fibril formation. a, b** Individual fits (solid lines) of normalized and averaged aggregation traces (crosses) of 3 µM Aβ42 in the presence of 0 (black), 10 (purple), 30 (cyan), 50 (yellow), 70 (green), and 100% (blue) rh Bri2 BRICHOS R221E monomer **a** and dimer **b** with the combined rate constants $\sqrt{k_n k_+}$ and $\sqrt{k_+ k_2}$ as free fitting parameters. The dependencies of the relative combined rate constants obtained from the fits (insets) reveal a strong effect of both rh Bri2 BRICHOS R221E species on secondary nucleation ($k_+ k_2$) but not on primary ($k_n k_+$) pathways. **c** Values for $r_{max}$ and $\tau_{1/2}$ (inset) extracted from the fitting of Aβ42 aggregation traces in the presence of different concentrations of rh Bri2 BRICHOS R221E species as shown in **a** and **b**. **d–f** Seeded aggregation traces of 3 µM Aβ42 with 0.6 µM preformed Aβ42 fibrils in the presence of 0 (black), 10 (purple), 30 (cyan), 50 (yellow), 70 (green), and 100% (blue) rh Bri2 BRICHOS R221E monomer **d** and dimer **e**. **f** Elongation rates ($k_+$) determined from highly seeded aggregation kinetics in **d** and **e**. **g–i** 5 µM Aβ42 was incubated with and without 50% molar ratio of rh Bri2 BRICHOS R221E monomers **g** or dimers **h** overnight at 37 °C. The samples were treated with goat anti-Bri2 BRICHOS antibody and a gold-labeled secondary antibody, and characterized by TEM. The scale bars are 100 nm. The arrows indicate the fibril ends. **i** Number of gold particles located within 30 nm of eight Aβ42 fibril ends after cofibrillation with rh Bri2 BRICHOS R221E monomers and dimers, respectively. The data are reported as means ± standard deviation. ***$p$ < 0.001.

dependent manner, and already at low concentrations fibril-end elongation is noticeably retarded (Fig. 3e, f, Supplementary Data 3). The rh Bri2 R221E monomer, in contrast, showed only slight effects on fibril-end elongation (Fig. 3d, f). This shows that both rh Bri2 BRICHOS R221E species reduce Aβ42 fibrillization via effects on surface catalysed secondary nucleation, while fibril-end elongation is only substantially affected by the rh Bri2 BRICHOS dimers.

The different effects implicate that monomers and dimers associate differently to the fibril ends. To study this, we used anti-Bri2 BRICHOS immunogold staining and transmission electron microscopy (TEM). We found that both the monomer and dimer bind abundantly and with similar densities to the Aβ42 fibril surfaces (Fig. 3g, h), which support the kinetic analyses and provide a basis for the effects of both rh Bri2 BRICHOS R221E

species on secondary nucleation. Further, the numbers of rh Bri2 BRICHOS R221E dimers that bind to Aβ42 fibril ends are higher than for monomers (Fig. 3i), which corroborates the results from kinetic analyses that predominantly the dimer modulates fibril-end elongation.

**Bri2 BRICHOS monomers efficiently prevent Aβ42 neurotoxicity.** Relative inhibition of Aβ42 secondary nucleation and elongation may be linked to the reduction of formation of toxic low-molecular weight species, as the number of new nucleation units is decreased by inhibiting secondary nucleation but increased by inhibiting elongation[19]. The results shown in Fig. 3f hence could have important implications for the expected amounts of toxic Aβ42 species formed in the presence of rh Bri2

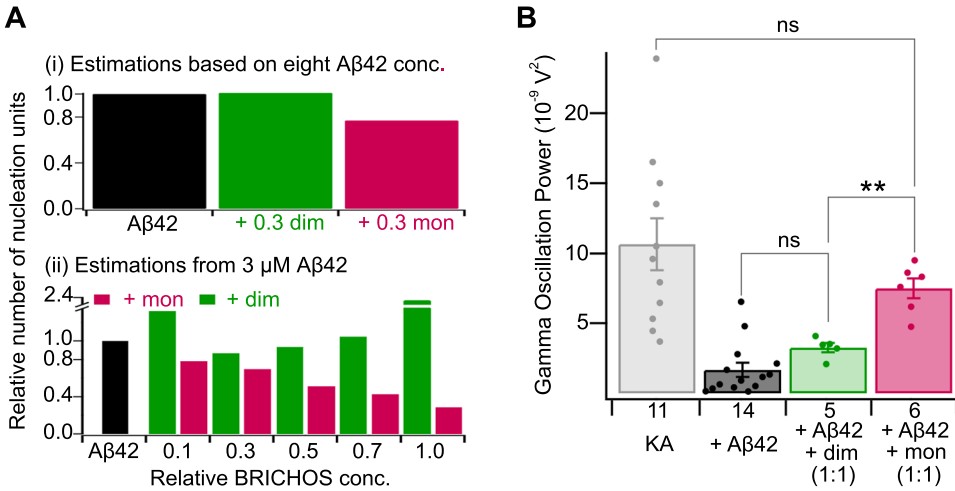

**Fig. 4 Effect of rh Bri2 BRICHOS R221E monomers and dimers on Aβ42 oligomer formation and neurotoxicity. a** Upper panel: relative numbers of Aβ42 nucleation units formed in the absence or presence of 30% molar ratio rh Bri2 BRICHOS R221E monomers and dimers estimated from the global fitting kinetics parameters derived from Supplementary Fig. 6 and the elongation rates ($k_+$) in Fig. 3f. Lower panel: the relative number of Aβ42 nucleation unit formed in the presence of different concentrations of rh Bri2 BRICHOS R221E monomers and dimers estimated from the individual fitting parameters derived from Fig. 3a, b and the elongation rates ($k_+$) from Fig. 3f. **b** Summary dot plot of mouse hippocampal γ oscillation power under control conditions (gray), after 15 min incubation with 50 nM Aβ42 (black), and after 15 min incubation with 50 nM Aβ42 + 50 nM rh Bri2 BRICHOS R221E monomers (red) or dimers (green). The numbers under the histograms denote the number of biological replicates, and the data are reported as means ± standard errors of the means. Control vs. Aβ42: $p < 0.0001$, ns, no significant difference, **$p < 0.01$.

BRICHOS R221E dimers and monomers, respectively. The amounts of new Aβ42 nucleation units were first calculated from the combined rate constants, determined from the global fit of different Aβ42 concentrations in the presence and absence of constant concentration of rh Bri2 BRICHOS R221E (Supplementary Fig. 6a–c), and the Aβ42 fibril elongation rate at 30% molar equivalent of different rh Bri2 BRICHOS R221E species determined from the seeded aggregation experiments (Fig. 3d–f). The results (Fig. 4a, upper panel) showed that generation of nucleation units is reduced by ca. 25% in the presence of 30% rh Bri2 BRICHOS R221E monomers, while the dimers had no effect. To investigate whether this inhibition effect is also active at higher relative Bri2 BRICHOS concentrations, we next estimated the generation of nucleation units from the combined rate constants obtained from the individual fits (Fig. 3a, b), which are afflicted with a larger uncertainty compared to the values obtained from the global fit, and the elongation rates (Fig. 3f). And indeed, the generation of nucleation units is reduced in a dose-dependent manner and up to 70% in the presence of rh Bri2 BRICHOS R221E monomers (Fig. 4a, lower panel). The results hence suggest that rh Bri2 BRICHOS R221E monomers reduce the formation of Aβ42 nucleation units, with an efficiency comparable to that of rh proSP-C BRICHOS[19], while the dimers do not.

To investigate whether the abilities of rh Bri2 BRICHOS R221E to reduce Aβ42-associated neurotoxicity correlates with the generation of nucleation units, we tested the efficacies of rh Bri2 BRICHOS monomers and dimers in preventing Aβ42-induced reduction of γ oscillations in mouse hippocampal slices. We found that addition of 50 nM rh Bri2 BRICHOS R221E monomer to 50 nM Aβ42 prevented Aβ-induced neurotoxicity and the power of γ oscillations reached the levels of non-treated controls (Fig. 4b). In contrast, addition of 50 nM rh Bri2 BRICHOS R221E dimer to 50 nM Aβ42 did not result in prevention of Aβ42-induced toxicity (Fig. 4b). This indicates that the monomers are efficient against Aβ42-induced toxicity, which supports the kinetics analysis that inhibition of secondary nucleation reduces the toxic Aβ42 oligomer formation. Noteworthily, increasing the concentration of the rh Bri2 BRICHOS

R221E dimer from 50 nM to 100 nM resulted in prevention of Aβ42-induced toxicity (Supplementary Fig. 10). These results show that the eventual outcome of inhibitors of Aβ42 fibril formation in terms of effects on neurotoxicity depends both on mechanism of action and molecular ratio relative to Aβ42.

## Discussion

Monomeric Bri2 BRICHOS species most efficiently prevent Aβ42-induced neurotoxicity, and hence their assembly into oligomers result in reduced anti-Aβ42 neurotoxicity. In this study, we designed the R221E Bri2 BRICHOS mutant that forms stable monomers and make rh wild-type Bri2 BRICHOS oligomers release monomers. Rh Bri2 BRICHOS R221E monomers efficiently alleviate Aβ42-induced neurotoxicity by selectively blocking secondary nucleation, which has previously been shown to constitute a main source of toxic species during Aβ42 aggregation. Moreover, the capacity of rh Bri2 BRICHOS R221E monomers to disassemble wild-type oligomers results in improved ability to delay Aβ42 fibrillization and, importantly, reduces Aβ42 toxicity to hippocampal slice preparations (Fig. 2).

Means of counteracting proteotoxicity include chaperone-mediated prevention of amyloid formation, disaggregation of pre-existing aggregates, and aggregate sequestration[4]. We rationalized our results in a schematic model to visualize the inhibition efficiencies of the small Bri2 BRICHOS R221E species on specific nucleation events, during Aβ42 fibrillization and potential sequestrations of the toxic species (Fig. 5a). The generation of new nucleation units promoted by secondary nucleation has been suggested to be linked to formation of small neurotoxic Aβ42 species and thus a specific prevention of secondary nucleation events may be beneficial against Aβ42-induced neurotoxicity[18,19]. Yet, a recent study indicated that also agents that result in an increase in the overall aggregation rate, predominantly caused by an enhanced secondary nucleation rate, can be beneficial to suppress Aβ42 toxicity provided that additional interactions take place[54]. While the detailed mechanism of Aβ42 caused toxicity is still under investigation, efficient toxicity modulators may be especially suited to specifically interact with

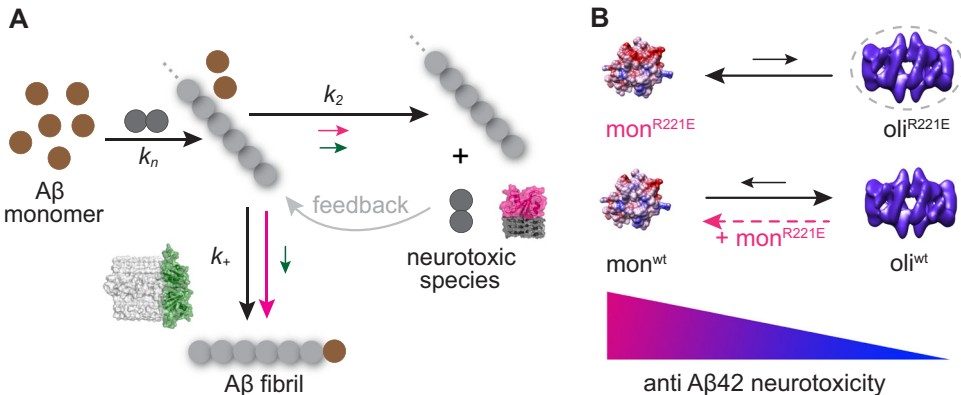

**Fig. 5 Model for potentiation of chaperone activity against Aβ42 neurotoxicity by shifting the Bri2 BRICHOS assembly state. a** Aβ42 forms fibrils via primary nucleation, elongation, and secondary nucleation, with rate constants $k_n$, $k_+$, and $k_2$, respectively[17]. While the Bri2 BRICHOS R221E dimer (green molecular model and green arrows) attenuates both $k_+$ and $k_2$, the monomer (red molecular model and red arrows) predominantly reduces $k_2$. Secondary nucleation catalyses the formation of new nucleation units, which acts as a positive feedback loop (grey arrow) for fibril formation, and this mechanism may be linked to enhanced generation of neurotoxic Aβ42 species[18,19]. Furthermore, the molecular size of the Bri2 BRICHOS monomer fits well to a single layer of β-structured Aβ42 molecules, which might be the structural element of neurotoxic Aβ42 species. On the contrary, the size of the dimer matches well the area of the cross section of fibril ends (PDB accession code 5KK3), potentially promoting attenuation of the fibril-end elongation rate. The structural properties together with the specific reduction of $k_2$ may thus make the Bri2 BRICHOS monomer most efficient in prevention of Aβ42-associated neurotoxicity. **b** Rh Bri2 BRICHOS R221E predominately forms monomers and smaller amounts of oligomers (hypothetical model in dashed circle), while rh wild-type (wt) Bri2 BRICHOS mainly assembles into high-molecular weight oligomers (electron microscopy data bank accession code EMD-3918) in equilibrium with monomers. Incubation of rh wt Bri2 BRICHOS oligomers with rh Bri2 BRICHOS R221E monomers destabilizes the oligomers and shifts the kinetic equilibrium toward the monomeric state, leading to an overall increased potency in preventing Aβ42 neurotoxicity (Fig. 2). This model provides thus a basis for understanding how the single point mutation R221E modulates the assembly state of Bri2 BRICHOS and thereby modulates its effects on Aβ42 fibril formation and enhances activities against Aβ42-associated neurotoxicity.

neurotoxic oligomeric Aβ42 species to shield surfaces that are available for aberrant interactions. These interactions may then hinder the underlying neurotoxic mechanism(s), such as direct binding to receptors or membrane destruction[4,55]. Interestingly, it has been shown that both intra- and extracellular chaperones, including clusterin, Hsp70, and αB-crystallin, can bind to oligomers of several different amyloidogenic peptides, including Aβ, and thereby induce their assembly into larger species that shield reactive surfaces[56]. It remains to be studied whether this mechanism applies to the BRICHOS domain as well, but electron microscopy data suggest that rh Bri2 BRICHOS monomers interact with oligomeric Aβ42 species[26]. In the current study, we found that the Bri2 BRICHOS R221E monomer predominantly attenuates the secondary nucleation rate, while the dimer substantially affects both the elongation and secondary nucleation rate (Figs. 3 and 5a). This causes a reduction of the generated number of nucleation units in the presence of Bri2 BRICHOS R221E monomers, but not dimers (Fig. 4a). The molecular size of the Bri2 BRICHOS R221E dimer matches well the surface of the cross sectional area of recently published Aβ42 fibril structures[57,58], providing a possible explanation why the Bri2 BRICHOS dimer, besides the secondary nucleation rate $k_2$, also efficiently attenuates the elongation rate $k_+$. The different specificity of the Bri2 BRICHOS dimer compared to the monomer also support that secondary nucleation and elongation events occur at distinct sites on Aβ42 aggregates, as recently reported[59]. While structural details about the soluble neurotoxic Aβ42 species are still missing, a β-structure state of toxic Aβ42 species/oligomers has been reported[55,60]. We observed that the molecular size of the Bri2 BRICHOS monomer fits well a single layer of β-structured Aβ42 molecules, which may build up neurotoxic Aβ42 species (Fig. 5a). Hence, the ability to efficiently reduce the generation of new nucleation units together with the well-matched molecular size for interactions with putative neurotoxic Aβ42 species potentially makes the Bri2 BRICHOS monomer most efficient in

preventing Aβ42-induced neurotoxicity to hippocampal γ oscillations (Fig. 4b).

For AD, aging is the main risk factor, but detailed underlying mechanisms are missing. Chaperone network decline during aging is considered to affect many aging-associated diseases[61]. In aging organisms, the balance between misfolded proteins and functional chaperones is disturbed[2]. Increased amounts of different Bri2 forms were observed in AD brains compared to the healthy controls[38]. Improved chaperone activity could be achieved by increasing the local concentration, but since chaperones are precisely balanced, overproduction of certain chaperone may result in disease[62]. Hence, the concept to augment the capacity of certain chaperone networks holds potential for preventing and treating pathologies associated with proteome deterioration[4]. The activity of α-crystallin was enhanced by addition of arginine hydrochloride or by methylglyoxal modification that changes the quaternary structure[63,64]. Chaperone activities can be selectively enhanced by grafting a peptide designed to bind a specific epitope onto a molecular chaperone, e.g., a grafted Hsp70 variants showed increased affinities for α-synuclein, Aβ42, and IAPP, respectively, but otherwise unaltered affinities for other substrates[7,65]. The activities of small Hsps, have been modulated by shifting their assembly states[66]. For example, at low temperature Hsp27 forms ~10 mers, whereas at elevated temperatures 22–30 mers were observed, concomitant with increased chaperone activity[67]. In contrast, mutations in αA- or αB-crystallin that result in the formation of larger aggregates decrease chaperone activity[68], while a bias toward monomeric substructure of α-crystallins was proposed to cause aberrant chaperone behaviour associated with protein deposition disease[69]. Disruption of a C-terminal tripeptide motif that binds to the αB-crystallin amyloid protein-binding surface impairs multimerization, shifts the equilibrium to smaller species, and results in enhanced chaperone activity[70]. Phosphorylation of the N-terminal domain of αB-crystallin results in destabilization of

larger oligomers (predominantly 24 mers), release of smaller species (predominantly 12 mers, 6 mers, and dimers) and concomitantly increased chaperone activity[71]. The chaperone activity of alkyl hydroperoxide reductase C from *Pseudomonas aeruginosa* was enhanced by a point-specific mutation that results in the formation of large oligomers, and the mutant confers heat tolerance to *E. coli*[72]. Recently, conjugation of a small-molecule inhibitor of α-synuclein aggregation to Hsp70 was shown to augment the anti-aggregation activity[8]. In our current study, augmentation of rh Bri2 BRICHOS activity against Aβ42-induced neurotoxicity is shown to occur as a result of formation of monomeric species from larger oligomers (Fig. 2). The results hence suggest a rational concept to enhance endogenous Bri2 BRICHOS activity against Aβ42-induced neurotoxicity.

Rational reprogramming of endogenous chaperone activity may find broad applicability in the fight against protein-misfolding diseases[1,5–8]. BRICHOS is a recently established molecular chaperone domain, which efficiently prevents fibrillar amyloid aggregation as well as its associated toxicity, as shown in vitro, in mouse hippocampal slice preparations, and in *Drosophila* models[19,23,26,27,32,34,35]. The dementia relevant Bri2 and its BBB permeable BRICHOS domain are associated with AD: Bri2 is produced in the CNS in the same cells as the Aβ precursor protein AβPP, colocalizes with senile plaques, interacts with Aβ in neurons, and increased amounts of different Bri2 forms have been found in human AD brains[38,39,48]. Further preclinical work that focus on the potential usefulness of Bri2 BRICHOS delivery and/or activation for prevention and treatment of AD should be pursued in relevant animal models.

## Methods

**Rh Bri2 BRICHOS R221E and Bri2 BRICHOS-AU1 preparation**. For generating rh Bri2 BRICHOS R221E the primers 5′-CACCTGGGTTTCTTTATTTAT-GAACTGTGTCATGACAAGGAAAC-3′ and 5′- GTTTCCTTGTCATGACA-CAGTTCATAAATAAAGAAACCCAGGTG-3′ were synthesized. With the wild-type NT*-Bri2 BRICHOS (corresponding to the solubility tag NT* followed by Bri2 residues 113–231[26,41]) plasmid as polymerase chain reaction (PCR) template Bri2 BRICHOS R221E (Addgene ID: 138134) was obtained with QuikChange II XL Site-Directed Mutagenesis Kit (Agilent, US), and the DNA sequence was confirmed (GATC Bioteq, Germany). As described previously[26,41], the proteins were expressed in SHuffle T7 competent *Escherichia coli* cells. Briefly, the cells were incubated at 30 °C in LB medium with 15 µg mL[−1] kanamycin until an OD$_{600 nm}$ ~0.9. For overnight expression, the temperature was lowered to 20 °C, and 0.5 mM Isopropyl β-D-1-thiogalactopyranoside was added. The induced cells were harvested by centrifugation (4 °C, 7000 × $g$) and the cell pellets were resuspended in 20 mM Tris pH 8.0. After sonicated (2 s on, 2 s off, 65% power) for 5 min on ice, the lysate was centrifuged (4 °C, 24,000 × $g$) for 30 min and the target protein was purified with a Ni-NTA column. To remove the His$_6$-NT* part, the target proteins were cleaved with thrombin (1:1000 enzyme to substrate, w/w) at 4 °C overnight and loaded over a second Ni-NTA column. Different rh Bri2 BRICHOS R221E species were isolated and analysed by Superdex 200 PG, 200 GL, or 75 PG columns (GE Healthcare, UK) using an ÄKTA system (GE Healthcare, UK).

For specific immunodetection, an AU1 tag (DTYRYI) or a mCherry domain was fused to the C-terminal of rh wild-type Bri2 BRICHOS by PCR amplification. The construct coding for His$_6$-NT*-thrombin cleavage site Bri2 BRICHOS wild-type-AU1 or His$_6$-NT*-thrombin cleavage site Bri2 BRICHOS wild-type-mCherry were expressed and purified with the same protocol as the one for Bri2 BRICHOS R221E. The rh wild-type Bri2 BRICHOS-AU1 or rh wild-type Bri2 BRICHOS-mCherry oligomers were isolated by Superdex 200 PG column (GE Healthcare, UK).

**ESI-MS of rh Bri2 BRICHOS R221E monomer**. Prior to ESI-MS analysis, rh Bri2 BRICHOS R221E monomers isolated by SEC were buffer exchanged using BioSpin microcentrifuge columns (BioRad, US) into 200 mM ammonium acetate pH 7.5. The final concentration of monomeric subunit was 20 µM. Spectra were recorded on a Waters Synapt G1 mass spectrometer (Waters, Milford, MA). Rh Bri2 BRICHOS R221E monomers were introduced using in-house produced gold-coated borosilicate capillaries. Instrument settings were: sample cone voltage 30 V, capillary voltage 1.5 V, collision trap voltage 50 V, extraction cone voltage 4 V, and transfer voltage 10 V. The source pressure was 7 mbar, trap gas was N$_2$ with a flow rate of 8 mL h[−1]. Data analysis was performed using Waters MassLynx 4.1 software.

**Circular dichroism, fluorescence, and aggregation analyses**. Circular dichroism spectra were recorded in 1 mm path length quartz cuvettes at 25 °C from 260 to 190 nm in an Aviv 410 Spectrometer (Lakewood, NJ, USA) with protein concentration of 12 µM. The bandwidth 1 nm, averaging time 0.3 s, wavelength step was 0.5 nm, and time constant 100 ms. The spectra shown are averages of three consecutive scans. One µM, calculated for the monomeric subunit, of different rh Bri2 BRICHOS R221E species in 20 mM Tris pH 8.0 were incubated at 25 °C with 2 µM bis-ANS (4,4′-Bis(phenylamino)-[1,1′-binaphthalene]-5,5′-disulfonic acid dipotassium salt) for 10 min, and the fluorescence emission spectra were recorded from 420 to 600 nm after excitation at 395 nm with an SLM-Aminco AB-2 spectrofluorimeter and thermostated cuvette holder (Thermo Spectronic, Waltham, MA, USA). Citrate synthase from porcine heart (Sigma-Aldrich, Germany) was diluted in 40 mM HEPES/KOH pH 7.5 to 600 nM and then equilibrated at 45 °C with and without different concentrations of various rh Bri2 BRICHOS R221E species. The aggregation kinetics were measured by reading the apparent increase in absorbance at 360 nm under quiescent conditions using a microplate reader (FLUOStar Galaxy from BMG Labtech, Offenberg, Germany).

**Rh Bri2 BRICHOS R221E incubation**. Rh Bri2 BRICHOS R221E oligomers, dimers, and monomers at 20 µM concentrations (referred to the monomeric subunits) were incubated in ThT assay buffer (20 mM sodium phosphate, pH 8.0 with 0.2 mM EDTA, and 0.02% NaN$_3$) at 37 °C. Samples were taken out after 0, 1, 4, and 24 h, and analysed for assembly states by sodium dodecyl sulfate–PAGE under reducing and non-reducing conditions. Ten µM rh Bri2 BRICHOS R221E oligomers and monomers were incubated in ThT assay buffer at 37 °C overnight and analyzed by native PAGE. Twenty µM rh Bri2 BRICHOS R221E dimers were also incubated in 20 mM sodium phosphate, pH 8.0 with 0.2 mM EDTA, and 0.02% NaN$_3$ at 37 °C overnight and analyzed by native PAGE.

**Aβ42 monomer preparation and ThT assay**. Recombinant Met-Aβ(1–42), here referred to as Aβ42, was produced in BL21*(DE3) pLysS *E. coli* (B strain) cells and purified by ion exchange[26]. The purified Aβ42 proteins were lyophilized and redissolved in 7 M Gdn-HCl, and the monomers were isolated in 20 mM sodium phosphate pH 8.0 with 0.2 mM EDTA and 0.02% NaN$_3$ by a Superdex 75 column (GE Healthcare, UK). The monomeric Aβ42 concentration was calculated using an extinction coefficient of 1,424 M[−1] cm[−1] for (A$_{280}$–A$_{300}$). For Aβ42 fibrillization kinetics analysis, 80 µL solution containing 10 µM ThT, 3 µM Aβ42 monomer, and different concentrations of rh Bri2 BRICHOS R221E monomer or dimer (all concentrations refer to the monomeric subunit) at molar ratios 0, 10, 30, 50, 70, and 100% (relative to Aβ42), were added to each well of half-area 96-well microplates with clear bottom (Corning Glass 3881, USA), and incubated at 37 °C under quiescent conditions. The ThT fluorescence was recorded using a 440 nm excitation filter and a 480 nm emission filter using a microplate reader (FLUOStar Galaxy from BMG Labtech, Offenberg, Germany). The aggregation kinetics of Aβ42 at different concentrations in the presence of 0.9 µM rh Bri2 BRICHOS R221E monomers and dimers were measured in the same manner. For Aβ42 seeds preparation, 3 µM Aβ42 monomer was incubated for ~20 h at 37 °C, and the fibrils were then sonicated in a water bath for 3 min. For seeding kinetics analysis of Aβ42 fibrillization, 80 µL solution containing 10 µM ThT, 3 µM Aβ42, different concentrations of rh Bri2 BRICHOS R221E monomers or dimers at 0, 10, 30, 50, 70, and 100%, and 0.6 µM Aβ42 seeds (calculated from the original Aβ42 monomer concentration) were added at 4 °C to each well in triplicate of 96-well microplates with clear bottom (Corning Glass 3881, USA) and immediately incubated at 37 °C under quiescent conditions. The elongation rate constant $k_+$ (see below) in the presence of rh Bri2 BRICHOS R221E monomers or dimers was calculated from the highly seeded experiments. The initial slope of the concave aggregation traces was determined by a linear fit of the first 25–30 min traces. For all the experiments, aggregation traces were normalized and averaged using 3–4 replicates, and data defining one dataset was recorded from the same plate.

**Coincubation of BRICHOS R221E monomer and wild-type oligomers**. Rh Bri2 BRICHOS R221E monomer (5 µM), rh wild-type Bri2 BRICHOS oligomer (5 µM), and the mixtures (rh Bri2 BRICHOS R221E monomer and rh wild-type Bri2 BRICHOS oligomer, 5 µM:5 µM or 9 µM:4.5 µM) were incubated at 37 °C overnight. Then the activities (at equal total rh Bri2 BRICHOS concentration) were tested against Aβ42 fibril formation by ThT assay, and for prevention of Aβ42 neurotoxicity by γ-oscillation measurements in hippocampal slices (see below). The samples were also analysed by native PAGE and western blotting using rabbit anti-AU1 antibody (1:1000, Abcam, UK).

**Analysis of Aβ42 aggregation kinetics**. Macroscopic aggregation profiles of Aβ42 in the presence of different concentrations of rh Bri2 BRICHOS R221E monomers or dimers were fitted to an empirical sigmoidal equation[73,74]:

$$F = F_0 + A/(1 + \exp[r_{max}(\tau_{1/2} - t)]) \qquad (1)$$

where $\tau_{1/2}$ is the aggregation half time, and $r_{max}$ the maximal growth rate, $A$ the amplitude, and $F_0$ the base value. By applying Eq. (1), the $\tau_{1/2}$ and $r_{max}$ were evaluated for aggregation traces with and without rh Bri2 BRICHOS R221E species.

The aggregation traces of the total fibril mass concentration, $M(t)$, is described by the following integrated rate law[19]

$$\frac{M(t)}{M(\infty)} = 1 - \left( \frac{B_+ + C_+}{B_+ + C_+ \times \exp(\kappa t)} \times \frac{B_- + C_+ \times \exp(\kappa t)}{B_- + C_+} \right)^{\frac{k_\infty^2}{\tilde{k}_\infty \kappa}} \times \exp(-k_\infty t)$$

(2)

where the additional coefficients are functions of $\lambda$ and $\kappa$:

$$C_\pm = \pm \lambda^2/2/\kappa^2$$
$$k_\infty = \sqrt{2\kappa^2/(n_2(n_2+1)) + 2\lambda^2/n_C}$$
$$\tilde{k}_\infty = \sqrt{k_\infty^2 - 4C_+C_-\kappa^2}$$
$$B_\pm = (k_\infty \pm \tilde{k}_\infty)/2/\kappa$$

which are two combinations of the microscopic rate constants by $\lambda = \sqrt{2 \times k_+ k_n \times m(0)^{n_C}}$ and $\kappa = \sqrt{2 \times k_+ k_2 \times m(0)^{n_2+1}}$. The microscopic rate constants $k_n$, $k_+$, and $k_2$ are the primary nucleation, elongation, and secondary nucleation rate constants, respectively, and the parameters $n_C$ and $n_2$ are the reaction orders for primary and secondary nucleation, respectively.

We identified the microscopic events that are inhibited by rh Bri2 BRICHOS R221E by applying Eq. (2) to describe the macroscopic aggregation profiles and comparing the microscopic rate constants $k_+ k_2$ and $k_n k_+$ in the absence and presence of BRICHOS required to describe the time evolution of Aβ42 fibrillization.

First the kinetic traces at different initial Aβ42 monomer concentration with constant rh Bri2 BRICHOS R221E concentration were fitted globally (Supplementary Fig. 5), where $\sqrt{k_n k_+}$ and $\sqrt{k_+ k_2}$ are constrained to the same value across all concentrations. The kinetic data at constant Aβ42 concentration with different rh Bri2 BRICHOS R221E concentrations were then globally analysed by applying this kinetic nucleation model (Fig. 3), where $\sqrt{k_n k_+}$ and $\sqrt{k_+ k_2}$ are free fitting parameters across all concentrations. We also performed a global fit of the kinetic data set at constant Aβ42 concentration and different rh Bri2 BRICHOS R221E concentrations, where the fit was constrained such that one fitting parameter was held to a constant value across all rh Bri2 BRICHOS R221E concentrations (Supplementary Fig. 6), while the second parameter was the only free parameter. This procedure results in that only one rate constant, i.e., $k_n$, $k_+$, or $k_2$, is the sole fitting parameter[73,74].

To illustrate the generation of nucleation units, according to the nucleation rate[19] $r_n(t)$ by $r_n(t) = k_n m(t)^{n_c} + k_2 M(t) m(t)^{n_2}$, we evaluated the time evolution of the rate of new fibrils generated from monomers. The number of nucleation units was calculated by the integration of the nucleation rate $r_n(t)$.

**Immunogold staining of Aβ42 fibrils and electron microscopy**. The Aβ42 monomer (5 μM) was incubated at 37 °C with 50% rh Bri2 BRICHOS R221E monomers or dimers for ~15 h, and the generated fibrils were collected by centrifugation for 1 h (22,000 × g, 4 °C). The fibrils were resuspended in 20 μL 1× Tris-buffered saline (TBS), and 2 μL were applied to formvar-coated nickel grids with ~5 min incubation. For blocking, the grids were incubated for 30 min in TBS containing 1% bovine serum albumin, then washed by TBS for 3 × 10 min. The grids were incubated overnight at 4 °C with primary antibody (goat anti-Bri2 BRICHOS antibody, 1:200 dilution), and washed again with TBS for 3 × 10 min. Finally the grids were incubated with 10 nm gold particles coupled secondary antibody (anti-goat IgG secondary antibody, 1:40 dilution, BBI Solutions, UK, EM. RAG10) at room temperature for 2 h, and washed with 1 × TBS for 5 × 10 min. For staining, 2.5% uranyl acetate (2 μL) was added to each grid with 20 s incubation, and excess solution was removed carefully. The grids were dried at room temperature for ~20 sec, and analysed by TEM (Jeol JEM2100F at 200 kV).

**NMR studies with rh Bri2 BRICHOS R221E species and Aβ40**. Lyophilized $^{15}$N-Aβ40 was dissolved in 10 mM NaOH to a concentration of 2 mg mL$^{-1}$ and sonicated for 2 min in a water-ice bath. The solution was diluted to reach a final Aβ concentration of 75 μM in 16 mM sodium–phosphate buffer, pH 7.4, 0.02% NaN$_3$, 0.2 mM EDTA, and 5% D$_2$O. $^1$H-$^{15}$N HSQC spectra were recorded at 8 °C on a 500 MHz Bruker spectrometer, equipped with a cryogenic probe, using 2048 × 128 complex points and 32 scans.

**Electrophysiological studies with rh Bri2 BRICHOS R221E**. All the experiments were carried out in accordance with the ethical permit granted by Norra Stockholm's Djurförsöksetiska Nämnd (dnr N45/13). C57BL/6 mice of either sex (postnatal days 14–23, supplied from Charles River, Germany) were used in the experiments. Before sacrificed, all the mice were anesthetized deeply using isofluorane.

The brain was dissected out and placed in modified ice-cold ACSF (artificial cerebrospinal fluid). The ACSF contained 80 mM NaCl, 24 mM NaHCO$_3$, 25 mM glucose, 1.25 mM NaH$_2$PO$_4$, 1 mM ascorbic acid, 3 mM NaPyruvate, 2.5 mM KCl, 4 mM MgCl$_2$, 0.5 mM CaCl$_2$, and 75 mM sucrose. Horizontal sections (350 μm thick) of the ventral hippocampi from both hemispheres were sliced with a Leica

VT1200S vibratome (Microsystems, Sweden). The sections were immediately transferred to a submerged incubation chamber containing standard ACSF: 124 mM NaCl, 30 mM NaHCO$_3$, 10 mM glucose, 1.25 mM NaH$_2$PO$_4$, 3.5 mM KCl, 1.5 mM MgCl$_2$, and 1.5 mM CaCl$_2$. The chamber was held at 34 °C for at least 20 min after dissection and it was subsequently cooled to room temperature (~22 °C) for a minimum of 40 min. Proteins (Aβ42, rh Bri2 BRICHOS R221E species as well as combinations) were first added to the incubation solution for 15 min, and then the slices were transferred to the interface-style recording chamber for extracellular recordings. During the incubation, slices were supplied continuously with carbogen gas (5% CO$_2$, 95% O$_2$) bubbled into the ACSF.

Recordings were performed with borosilicate glass microelectrodes in hippocampal area CA3, pulled to a resistance of 3–5 MΩ. Local field potentials (LFP) were recorded at 32 °C in an interface-type chamber (perfusion rate 4.5 mL per minute) using microelectrodes filled with ACSF placed in stratum pyramidale. LFP γ-oscillations were elicited by kainic acid (100 nM, Tocris). The oscillations were stabilized for 20 min before any recordings. No Aβ42, rh Bri2 BRICHOS R221E species, or combinations thereof were present in the recording chamber either during γ oscillations stabilization, or during electrophysiological recordings. The interface chamber recording solution contained 124 mM NaCl, 30 mM NaHCO$_3$, 10 mM glucose, 1.25 mM NaH$_2$PO$_4$, 3.5 mM KCl, 1.5 mM MgCl$_2$, and 1.5 mM CaCl$_2$.

Interface chamber LFP recordings were carried out by a 4-channel amplifier/signal conditioner M102 amplifier (Electronics lab, University of Cologne, Germany). The signals were sampled at 10 kHz, conditioned using a Hum Bug 50 Hz noise eliminator (LFP signals only; Quest Scientific, North Vancouver, BC, Canada), software low-pass filtered at 1 kHz, digitized, and stored using a Digidata 1322 A and Clampex 9.6 programs (Molecular Devices, CA, USA). Power spectral density plots (from 60 s long LFP recordings) were calculated using Axograph X (Kagi, Berkeley, CA, USA) in averaged Fourier segments of 8192 points. Oscillation power was from the integration of the power spectral density from 20 to 80 Hz.

**Statistics and reproducibility**. The electrophysiology data are presented as means ± standard errors of the means. The Student's *t*-test (unpaired) was used for statistical analysis. All experiments were performed with parallel controls from the same animal/preparation, and the number of biological replicates are given in the figure legends. Significance levels are *$p < 0.05$; **$p < 0.01$; ***$p < 0.001$. The ThT fluorescence data are presented as means ± standard deviation, and the aggregation traces are averaged from 3–4 replicates.

**Reporting summary**. Further information on research design is available in the Nature Research Reporting Summary linked to this article.

## Data availability
All data and materials related to this paper are available from J.J. (janne.johansson@ki.se). Plasmid Bri2 BRICHOS R221E has been deposited at Addgene (Addgene ID: 138134).

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

## Acknowledgements

This study was supported by the Swedish Research Council (J.J.), the Center for Innovative Medicine (CIMED; J.J.), the Olle Engkvists Stiftelse (G.C.), the Swedish Alzheimer foundation (G.C. and A.F.), the Swedish Brain Foundation (A.F.), the Åhlén-stiftelsens (G.C., A.A. and A.F.), Petrus and Augusta Hedlunds Stiftelse (G.C.), the Stiftelsen för Gamla Tjänarinnor (G.C., A.A. and H.B.), Instruct R&D pilot project grant APPID 272 (A.A.), the Stohnes Foundation (A.A.), Loo and Hans Osterman Foundation (G.C., A.A. and Y.A.T.), Geriatric Diseases Foundation at Karolinska Institutet (G.C., A.A., Y.A.T. and A.F.), Magnus Bergvall foundation (A.A. and H.B.), and the FLPP lzp-2018/1-0275 project support (H.B.).

## Author contributions

G.C., Y.A.T., S.T., H.E.N., A.L., X.Z. and M.L. performed experiments. G.C. and A.A. performed kinetic analyses. G.C., Y.A.T., A.A., S.T., P.N., H.H., A.F., H.B. and J.J. analyzed the data. J.J. conceived and supervised the study. G.C., A.A. and J.J. wrote the paper. All authors discussed the results and commented on the manuscript.

## Competing interests

The authors declare no competing interests.
