## [Peer Review File · Communications Biology]

Reviewers' comments:

Reviewer #1 (Remarks to the Author):

Johansson and co-workers extend the "BRICHOS saga" by cleverly designing a mutant with significantly reduced oligomerization tendencies, leading to much higher levels of monomer and dimer. Monomer and dimer are stable when purified by gel filtration and show remarkable but different effects on A β aggregation and cytotoxicity, which the authors are able to rationalize based on known A β fibril and BRICHOS monomer structures. Overall this is a very well constructed and thoroughly argued piece of work which sheds important new light on a topic of increasing relevance, namely the role of chaperones in controlling and reducing formation of neurotoxic oligomeric protein species.

I only have a few minor issues for the authors to address:

1. Fig. S3D: It is a bit difficult to read this graph. There is a change in signal with oligomers but less with dimers and none with oligomers. I would associate aggregation with an increase in turbidity, i.e. absorbance, but that is counter what is shown. Please clarify.

2. The authors claim that the wt-R221E "mixture attenuated Ab42-induced reduction of γ oscillation power more potently than 50 nM of rh wt Bri2 BRICHOS oligomers pre-incubated in the same way as the mixture" but the mix does not have higher values than the other two BRICHOS samples. Please clarify. Furthermore, I don't understand the relationship between Fig. 2AB and Fig. S4 which seems to be data accumulated under the same conditions but with different results. The labelling in the graph in Fig. S4 is also inconsistent with labelling in Fig. 2A (Ab42 is mentioned twice in Fig. S4 which is confusing). Further, mutant monomer seems to be performing worse than wt oligomer in terms of increasing gamma oscillation power compared to Ab42 alone, inconsistent with the text.

3. The supplementary figures are not shown in the right sequence (e.g. Fig. S8 appears very early in the text).

Reviewer #2 (Remarks to the Author):

Title: it has to be clear that the augmentation of chaperone activity against A β in this paper is in vitro.

Abstract: What main source of toxic A β species does R221E block? Or does it only work by reducing oligomerization of wild type BRICHOS?

Figure 1 . What happens with longer incubations? Have you ever looked for recovery of the gamma oscillation response?

Wt monomers dimerize, mutant monomers do not. Since the monomers are most effective in suppressing neurotoxicity and the dimers suppress fibril formation while the oligomers inhibit non fibrillary aggregation does reduction and alkylation of rh BRICHOS change the properties of the molecule with respect to these activities.

Figure 1

In 1C while it appears that the 1:1 mixture yields more immuno-reactive monomer than in the 2:1 mix, the coomassie stained gel looks like there is more monomer in the 2:1. Does that come from the mutant, which seems unlikely since there is no monomer in either mutant monomer lane? Are you saying that the mutant converts wt to monomer via dissociation and that the stoichiometry of the effect is better at 1:1 than 2:1? It also appears that the other species are equal or more prominent in the 2:1 mix. Is the dissociation of the oligomers sequential with monomers the last product? This raises the question of mass balance. Is all the starting material accounted for in the final reaction? Please discuss.

1E: It is unfortunate that you showed only a single time point in the incubation of the mutant monomer. This a perfect experimental setting to definitively determine what the interaction is between

R221E and wt protein by two color fluorescence size analysis methods and how it might change with time (or not).

Figure 2 is a problem for me. Panel A: the differences between controls and wt oligomer are significant and those between controls and mutant oligomers and mixture are not. Are the differences between wt and mut and mix significantly different, if not why not? Further the error bars for wt oligomers are much smaller than other experiments and there are no error bars for the controls. Is the smaller sd real? An artifactually small SD will enhance the statistical difference of the analysis. I found this confusing. This requires some explanation or discussion. Did you use any other assays of cytotoxicity? Is the A β effect based on cell killing or some other mechanism? Do the slices recover in time (or how long after a washout at 3 hrs does the pathway recover?).

You should probably use a more recent reference for γ oscillations, e.g.

Gamma frequency entrainment attenuates amyloid load and modifies microglia. Iaccarino HF, Singer AC, Martorell AJ, Rudenko A, Gao F, Gillingham TZ, Mathys H, Seo J, Kritskiy O, Abdurrob F, Adaikkan C, Canter RG, Rueda R, Brown EN, Boyden ES, Tsai LH. *Nature*. 2016 Dec 7;540(7632):230-235.

Multi-sensory Gamma Stimulation Ameliorates Alzheimer's-Associated Pathology and Improves Cognition. Martorell AJ, Paulson AL, Suk HJ, Abdurrob F, Drummond GT, Guan W, Young JZ, Kim DN, Kritskiy O, Barker SJ, Mangena V, Prince SM, Brown EN, Chung K, Boyden ES, Singer AC, Tsai LH. *Cell*. 2019 Apr 4; 177(2):256-271

With respect to examining the nature of the interaction between wt and R221E BRICHOS responsible for the activity in the hippocampal slice experiments it would have been preferable to examine the monomer oligomer relationships and the nature of the R221E molecules interacting with A β by carrying out a parallel incubation in the absence of tissue, and running native and non-native westerns with both antibodies with and without cross linking or even better with the BRICHOS labeled with one fluor and A β with another (non-overlapping).

In the experiments shown in figure 3 do you ever measure the amount of A β remaining in the supernatant fluid as non-fibrillar aggregates, i.e. whether BRICHOS allows the reaction to supersaturate for A β ? Do you ever cross-link the reaction mixes and assay what sized molecular species contains both proteins by western? This might get at the nature of the intermolecular interactions, while CFS experiments could give you single molecule, rather than averaged results.

General:

Mutant molecule oligomers suppress heat induced citrate synthase aggregation. This suggests that they function more like clusterin than other well characterized chaperones. Any comments given the prior observations of Wilson and Chiti regarding clusterin functioning by inducing the production of large, non-toxic amorphous aggregates?

Does any mutation in BRICHOS have the same effect as R221E? Have you tested other mutations? For R221E, What is the binding site? What is the K_d? What is the stoichiometry? For both A β and wt BRICHOS?

If you are proposing this as a therapeutic, I assume you think it will work through enhancing the effect of naturally produced Bri2, i.e. generating more monomer, is there any evidence that Bri2 production is increased in the presence of human neurodegenerative diseases or the relevant mouse models? Or do you think it can have some effect on its own?

REVIEWERS'S COMMENTS and **RESPONSE**

Dear Dr. Barabas,

Thank you for the positive news and thorough review of our manuscript. We have now responded to the comments raised by the reviewers and revised the manuscript accordingly, including the addition of more experimental data. Please see below for a detailed reply to the reviewers' points.

Reviewer #1 (Remarks to the Author):

Johansson and co-workers extend the "BRICHOS saga" by cleverly designing a mutant with significantly reduced oligomerization tendencies, leading to much higher levels of monomer and dimer. Monomer and dimer are stable when purified by gel filtration and show remarkable but different effects on Abeta aggregation and cytotoxicity, which the authors are able to rationalize based on known Abeta fibril and BRICHOS monomer structures. Overall this is a very well constructed and thoroughly argued piece of work which sheds important new light on a topic of increasing relevance, namely the role of chaperones in controlling and reducing formation of neurotoxic oligomeric protein species.

I only have a few minor issues for the authors to address:

1. Fig. S3D: It is a bit difficult to read this graph. There is a change in signal with oligomers but less with dimers and none with monomers. I would associate aggregation with an increase in turbidity, i.e. absorbance, but that is counter what is shown. Please clarify.

Reply: We are sorry for the unclear description of what is shown in the graph. The final absorbance intensity after aggregation of thermally destabilized citrate synthase for 1h in the presence of increasing concentrations of Bri2 BRICHOS R221E monomers, dimers and oligomers is shown. We have now clarified this in the figure legend.

2. The authors claim that the wt-R221E "mixture attenuated Ab42-induced reduction of g oscillation power more potently than 50 nM of rh wt Bri2 BRICHOS oligomers pre-incubated in the same way as the mixture" but the mix does not have higher values than the other two BRICHOS samples. Please clarify.

Reply: That sentence refers only to the 50 nM of rh wt Bri2 BRICHOS oligomers pre-incubated in the same way as the mixture not to the mutant oligomers. We have remade Figure 2 and better explained the results in the text. See also the reply to the next point and the reply to reviewer #2 below.

Furthermore, I don't understand the relationship between Fig. 2AB and Fig. S4 which seems to be data accumulated under the same conditions but with different results. The labelling in the graph in Fig. S4 is also inconsistent with labelling in Fig. 2A (Ab42 is mentioned twice in Fig. S4 which is confusing). Further, mutant monomer seems to be performing worse than wt oligomer in terms of increasing gamma oscillation power compared to Ab42 alone, inconsistent with the text.

Reply: We agree that the description of the results shown in Figures 2 and S4 were unclear. We have now remade the figure and included the data from Supplementary Figure S4, which showed data for half the concentrations (25 nM) used in Figure 2 (50 nM) in order to show the respective effects of the 25 nM rh wt Bri2 BRICHOS oligomers and 25 nM mutant monomers, ie the components used in the mixture.

3. The supplementary figures are not shown in the right sequence (e.g. Fig. S8 appears very early in the text).

Reply: Thanks for spotting this error, we have now renumbered the supplementary figures so that they appear in the right order.

Reviewer #2 (Remarks to the Author):

Title: it has to be clear that the augmentation of chaperone activity against A β in this paper is *in vitro*.

Reply: We have added "in mouse hippocampus *in vitro*" to the title.

Abstract: What main source of toxic A β species does R221E block? Or does it only work by reducing oligomerization of wild type BRICHOS?

Reply: The R221E mutant predominantly blocks the secondary nucleation reaction during A β 42 fibril formation (Figures 3 and 4), which according to literature data is the main source of toxic species during A β 42 fibril formation (see Cohen et al Nat Struct Mol Biol 2015, 22, 207).

Figure 1. What happens with longer incubations? Have you ever looked for recovery of the gamma oscillation response?

Reply: Rh Bri2 BRICHOS can indeed rescue A β 42 induced reduction of gamma oscillations, see figure 1 below. This is, however, a topic on its own and outside the scope of the present manuscript.

Figure 1. Time course of gamma power in control conditions (black), in slices pre-incubated only with 50 nM Aβ42 (red) and effect of 1 μM rh Bri2 BRICHOS (green) application to slices pre-incubated with 50 nM Aβ42. YAT, GC et al, unpublished data.

Wt monomers dimerize, mutant monomers do not. Since the monomers are most effective in suppressing neurotoxicity and the dimers suppress fibril formation while the oligomers inhibit non fibrillary aggregation does reduction and alkylation of rh BRICHOS change the properties of the molecule with respect to these activities.

Reply: We have not performed the suggested experiment, since we have preliminary data showing that mild reduction of the intramolecular disulphide in rh Bri2 BRICHOS monomer results in oligomerization (probably secondary to exposure of hydrophobic regions) and formation of intermolecular disulphide(s). Moreover, mutation of both Cys residues leads to an unstable protein that is partly degraded during expression and purification. Further studies of the reactivities of the two strictly conserved Cys in BRICHOS will be a topic for future studies.

1C: while it appears that the 1:1 mixture yields more immuno-reactive monomer than in the 2:1 mix, the coomassie stained gel looks like there is more monomer in the 2:1. Does that come from the mutant, which seems unlikely since there is no monomer in either mutant monomer lane? Are you saying that the mutant converts wt to monomer via dissociation and that the stoichiometry of the effect is better at 1:1 than 2:1? It also appears that the other species are equal or more prominent in the 2:1 mix. Is the dissociation of the oligomers sequential with monomers the last product? This raises the question of mass balance. Is all the starting material accounted for in the final reaction? Please discuss.

Reply: We suppose that Figure 2C is referred to. Please note that the Western blot and Coomassie staining are not directly comparable since the former detects only wt protein (ie containing the AU tag) while Coomassie stains both wt and mutant protein. This is why the mutant monomer is detected by Coomassie but absent in the Western blot. The inclusion of the AU tag in the wt protein was necessary in order to allow differentiation between mutant monomers and released wt monomers. We agree that the amounts of wt monomers for 1:1 and 2:1 ratios between mutant monomer and wt oligomers are counterintuitive. We therefore performed additional experiments in which fluorescently labeled wt oligomers were titrated with mutant monomers and the release of wt monomers was quantified for different ratios between mutant monomer and wt oligomers. This showed that the release of wt monomers shows a dose dependence relative to the amounts of mutant monomers present. These data are now presented in Supplementary Figure 5 and on page 7.

1E: It is unfortunate that you showed only a single time point in the incubation of the mutant monomer. This a perfect experimental setting to definitively determine what the interaction is between R221E and wt protein by two color fluorescence size analysis methods and how it might change with time (or not).

Reply: We are unsecure on what Figure the reviewer refers to. Figure 1E shows a native PAGE for mutant oligomers. But we have now performed experiments with fluorescently labeled wt oligomers and mutant monomers, see reply above and Supplementary Figure 5.

Figure 2 is a problem for me. Panel A: the differences between controls and wt oligomer are significant and those between controls and mutant oligomers and mixture are not. Are the differences between wt and mut and mix significantly different, if not why not? Further the error bars for wt oligomers are much smaller than other experiments and there are no error bars for the controls. Is the smaller sd real? An artifactually small SD will enhance the statistical difference of the analysis. I found this confusing. This requires some explanation or discussion. Did you use any other assays of cytotoxicity? Is the A β effect based on cell killing or some other mechanism? Do the slices recover in time (or how long after a washout at 3 hrs does the pathway recover?).

Reply: We apologize for not having explained the results in Figure 2 and old Supplementary Figure S4 in a clear manner. We have now revised the Figure showing the results of combinations of A β 42 and different rh Bri2 BRICHOS species on gamma oscillations. The experiments involving the mixture of mutant monomer and wt oligomer and the oligomers alone were performed during different time periods. The values for the control gamma oscillations differ between these two sets of experiments and the values for the gamma oscillations after different incubations differ in a corresponding manner (see the respective power spectra in Fig. 2A). The

differences observed in our different rounds could thus be due to seasonal variations that have been observed to have an impact on brain physiology (please see Sahar Farajnia et al 2014 Seasonal induction of GABAergic excitation in the central mammalian clock, PNAS 111:9627; Hofman and Swaab 1993 Diurnal and Seasonal Rhythms of Neuronal Activity in the Suprachiasmatic Nucleus of Humans, J Biol Rhythms 8:283; and Weiner et al 1992 Circadian and seasonal rhythms of 5-HT receptor subtypes, membrane anisotropy and 5-HT release in hippocampus and cortex of the rat, Neurochem Int 21:7). This makes it problematic to compare the results for the effects of adding the Bri2 BRICHOS mixture to A β 42 to those obtained for the oligomers added to A β 42. This is the reason why we did not include a direct statistical comparison of these two sets of experiments. In the revised Figure 2 it is now clear that the experiments were performed separately, and the two respective control groups are now shown.

We have not performed studies on cell death or other measures of toxicity since acute A β neurotoxicity is not thought to be mediated by direct cytotoxicity but rather by effects on synapse function and neuronal network activity. Please note that the A β 42 concentrations used in our hippocampal slice studies (50 nM) are close to the physiological A β 42 concentrations but smaller than the microM concentrations used in cell toxicity assays. Regarding recovery, Figure 1 above demonstrate that 15 min A β effects last more than 2h.

You should probably use a more recent reference for γ oscillations, e.g.

Gamma frequency entrainment attenuates amyloid load and modifies microglia.

Iaccarino HF, Singer AC, Martorell AJ, Rudenko A, Gao F, Gillingham TZ, Mathys H, Seo J, Kritskiy O, Abdurrob F, Adaikkan C, Canter RG, Rueda R, Brown EN, Boyden ES, Tsai LH. Nature. 2016 Dec 7;540(7632):230-235.

Multi-sensory Gamma Stimulation Ameliorates Alzheimer's-Associated Pathology and Improves Cognition. Martorell AJ, Paulson AL, Suk HJ, Abdurrob F, Drummond GT, Guan W, Young JZ, Kim DN, Kritskiy O, Barker SJ, Mangena V, Prince SM, Brown EN, Chung K, Boyden ES, Singer AC, Tsai LH. Cell. 2019 Apr 4; 177(2):256-271

Reply: Thank you, these references have now been included.

With respect to examining the nature of the interaction between wt and R221E BRICHOS responsible for the activity In the hippocampal slice experiments it would have been preferable to examine the monomer oligomer relationships and the nature of the R221E molecules interacting with A β by carrying out a parallel incubation in the absence of tissue, and running native and non-native westerns with both antibodies with and without cross linking or even better with the BRICHOS labeled with one fluor and A β with another (non-overlapping).

Reply: Fluorescently labeled A β or BRICHOS are not suited for the hippocampal slice experiments since the fluorophores may affect the results, which will make it difficult to compare to data from non-labeled recombinant A β 42.

In the experiments shown in figure 3 do you ever measure the amount of A β remaining in the supernatant fluid as non-fibrillar aggregates, i.e. whether BRICHOS allows the reaction to supersaturate for A β ? Do you ever cross-link the reaction mixes and assay what sized molecular species contains both proteins by western? This might get at the nature of the intermolecular interactions, while CFS experiments could give you single molecule, rather than averaged results.

Reply: We have determined the amount of A β 42 that goes into the fibrils with and without BRICHOS present, and find that essentially all A β 42 is incorporated into the fibrils. These results are now included in Supplementary Figure 7 and described on p. 8.

General:

Mutant molecule oligomers suppress heat induced citrate synthase aggregation. This suggests that they function more like clusterin than other well characterized chaperones. Any comments given the prior observations of Wilson and Chiti regarding clusterin functioning by inducing the production of large, non-toxic amorphous aggregates?

Reply: We thank the reviewer for the suggestion, and now discuss this on p. 12.

Does any mutation in BRICHOS have the same effect as R221E? Have you tested other mutations?

Reply: We have investigated several mutations, and found that so far R221E is the most suitable one for maintaining a monomeric state. The double mutants D148NR221E and F217RR221E did not show obvious changes to the oligomerization profile compared to R221E (figure 2).

Figure 2. Size exclusion chromatography of rh Bri2 BRICHOS R221E, Bri2 BRICHOS D148NR221E, and Bri2 BRICHOS F217RR221E.

For R221E, What is the binding site? What is the Kd? What is the stoichiometry? For both A β and wt BRICHOS?

Reply: We are working on getting numbers on these parameters but do not have results to present at the present stage. We have included NMR experiments on monomeric A β 40 and rh Bri2 BRICHOS R221E (Supplementary Figure 9), which show that there is no detectable interaction between any specific region of monomeric A β and the BRICHOS mutant.

If you are proposing this as a therapeutic, I assume you think it will work through enhancing the effect of naturally produced Bri2, i.e. generating more monomer, is there any evidence that Bri2 production is increased in the presence of human neurodegenerative diseases or the relevant mouse models? Or do you think it can have some effect on its own?

Reply: These are questions that we currently are addressing. Del Campo et al 2014, Neurobiol Aging, detected increased amounts of immunoreactive Bri2 in Alzheimer brains compared to controls (mentioned in the Introduction) but data for mouse models are not available.

REVIEWERS' COMMENTS:

Reviewer #1 (Remarks to the Author):

The authors have satisfactorily addressed my (modest) concerns and I have no further issues with their manuscript.

Reviewer #2 (Remarks to the Author):

Thank you for responding to my queries. It is unfortunate that most of the responses cited work in progress or planned rather than data that would be appropriate for enhancing the interpretation of the present manuscript in the context of the in vivo impact of chaperones on the pathogenesis of AD. A summary of the findings in the context of what has been shown for Bri proteins in AD in vivo rather than the theoretical in vitro models of Cohen et al would have been helpful for the more biologically oriented reader.

REVIEWERS' COMMENTS:

Reviewer #1 (Remarks to the Author):

The authors have satisfactorily addressed my (modest) concerns and I have no further issues with their manuscript.

Reply: thank you.

Reviewer #2 (Remarks to the Author):

Thank you for responding to my queries. It is unfortunate that most of the responses cited work in progress or planned rather than data that would be appropriate for enhancing the interpretation of the present manuscript in the context of the in vivo impact of chaperones on the pathogenesis of AD. A summary of the findings in the context of what has been shown for Bri proteins in AD in vivo rather than the theoretical in vitro models of Cohen et al would have been helpful for the more biologically oriented reader.

Reply: thank you for the relevant suggestion. We have added one last paragraph in the discussion that summarizes literature data on Bri2 in relation to AD, and also highlights the general importance of chaperones for AD and other protein misfolding diseases.